# Untwisted restacking of two-dimensional metal-organic framework nanosheets for highly selective isomer separations

Ze-Rong Tao[1], Jian-Xiang Wu[1], Ying-Jie Zhao[2], Ming Xu[1], Wen-Qi Tang[1], Qing-Hua Zhang[3], Lin Gu [3], Da-Huan Liu[2] & Zhi-Yuan Gu[1]

The stacking between nanosheets is an intriguing and inevitable phenomenon in the chemistry of nano-interfaces. Two-dimensional metal-organic framework nanosheets are an emerging type of nanosheets with ultrathin and porous features, which have high potential in separation applications. Here, the stacking between single-layer metal-organic framework nanosheets is revealed to show three representative conformations with tilted angles of 8°, 14°, and 30° for Zr-1, 3, 5-(4-carboxylphenyl)-benzene framework as an example. Efficient untwisted stacking strategy by simple heating is proposed. A detailed structural analysis of stacking modes reveals the creation of highly ordered sub-nanometer micropores in the interspacing of untwisted nano-layers, yielding a high-resolution separator for the pair of *para-/meta*-isomers over the twisted counterparts and commercial HP-5MS and VF-WAXMS columns. This general method is proven by additional nanosheet examples and supported by Grand Canonical Monte Carlo simulation. This finding will provide a synthetic route in the rational design of functionalities in two-dimensional metal-organic framework nanosheet.

[1] Jiangsu Key Laboratory of Biofunctional Materials, Jiangsu Collaborative Innovation Center of Biomedical Functional Materials, College of Chemistry and Materials Science, Nanjing Normal University, 210023 Nanjing, China. [2] State Key Laboratory of Organic–Inorganic Composites, Beijing Advanced Innovation Center for Soft Matter Science and Engineering, Beijing University of Chemical Technology, 100029 Beijing, China. [3] Institute of Physics, Chinese Academy of Sciences, 100190 Beijing, China. Correspondence and requests for materials should be addressed to Z.-Y.G. (email: guzhiyuan@njnu.edu.cn)

Ordering and disordering in the molecular self-organization system is the common phenomena, while random distribution and re-ordering in supermolecular stacking system, especially within sub-nanometer between nanosheets is rarely explored[1–3]. Two-dimensional (2-D) metal-organic framework (MOFs) nanosheets as a novel category of 2-D materials are attracting increasing interest due to their ultrathin thickness, highly accessible functional surface, and excellent electronic or photonic properties[4–6], which make them potential in catalysis[7], energy storage[8], sample preparation[9], fluorescent sensing[10,11], and enzyme inhibition[12]. Building or destroying the covalent or coordination bonds between adjacent MOF layers is efficient way for the construction or deconstruction of frameworks. However, the tuning of weak interactions between MOF nanosheets is largely unexplored although the tuning of stacking modes in conventional nanosheets, such as graphene[13] and $MoS_2$[14] etc, is of great potential in the separation applications.

The separation and detection of structural isomers of substituted benzene derivatives mixtures (such as ortho-xylene (oX), meta-xylene (mX), para-xylene (pX), and ethylbenzene) is a difficult and significant process either in petroleum industry or air monitoring, since they are important raw chemicals and environmental pollutants[15,16]. Such challenges arise from their extremely similar physical and chemical properties, including molecular shape and polarity. Energy-efficient adsorbents-based separations, such as membrane separation[17–19], breakthrough[20,21], and chromatographic separation[22–25] are promising in state-of-art industrial applications compared with crystallization and distillation approaches. The high resolution chromatographic technique is capable of distinguishing molecular sizes and shapes and providing energetic information, which is an efficient tool in the materials screening, as well as analytical practice in real sample analysis. The chromatographic separation shows superior advantages on the instant determination of selectivity over other separation techniques, which require calibration and tedious calculation. The selectivity is a direct indicator of pore characteristics and of practical in industrial separation, such as para-isomer preference. Recently, 2-D MOF nanosheets as ultrapermeable and highly selective membranes for sieving small gas molecules, such as $H_2$, $CO_2$, $CH_4$ have been reported[26–28] because of their nanometersized thickness along with the large amount of regular pore arrays in the plane providing precise recognition at molecular level. Nevertheless, to the best of our knowledge, 2-D MOF nanosheets have not been used as stationary phase in chromatography to date.

Here, to the best of our knowledge, we first decrypt the stacking modes between adjacent MOF nanosheets with Zr-BTB-FA (BTB = 1, 3, 5-(4-carboxylphenyl)-benzene, FA = formic acid) as an example (Fig. 1a). Three specific angles of 8°, 14°, and 30° between tilted nanosheets stacking of Zr-BTB-FA are observed on high-angle annular dark field (HAADF) images. Untwisted stacking is achieved through preheating strategy to get highly ordered nanosheets (Fig. 1b). Powder X-ray diffraction (PXRD), $N_2$ adsorption, proton nuclear magnetic resonance (H$^1$NMR), diffuse reflectance infrared fourier transform spectroscopy (DRIFTS), and atomic force microscopy (AFM) are utilized to demonstrate the untwisted stacking nanosheets with microporosity and multiple layer structures. The first fabrication of MOF nanosheets-coated gas chromatographic (GC) capillary columns is achieved to show that untwisted restacking greatly enhances the separation efficiency, not only superior to twisted stacking nanosheets but also much better even than the commercial HP-5MS and VF-WAXMS columns for para/meta isomer separation (Fig. 1c). Meanwhile, these general untwisted stacking strategy is proved by other two Zr-BTB MOF nanosheets with different functionalities (Zr-BTB-BA, Zr-BTB-PABA,

BA = benzoic acid, PABA = para-aminobenzoic acid). Grand Canonical Monte Carlo (GCMC) simulation locates the highly selective cavity and confirms the para-selectivity, which is in consistence with the enthalpy measurements. This work sheds light on the frontiers of stacking phenomenon in the state-of-art nanosheets and opens a new window in the creation of highly selective porous materials.

## Results and discussion

**Twisted and untwisted stacking of Zr-BTB nanosheets.** Here, we chose surfactant-free bottom-up approach to synthesize 2-D MOF nanosheets and employed formic acid as a saturated modulator to overcome high surface energy and suppress the growth along the vertical direction[29]. The ultrathin 2-D Zr-BTB-FA nanosheets with 6-connected $Zr_6$ clusters and 3-connected BTB ligands were successfully synthesized. To investigate the stacking modes, the 2-D Zr-BTB-FA was first dried under ambient conditions for 24 h, then analyzed by HAADF. In the HAADF images, 2-D kgd nets with white spots representing $Zr_6$ clusters were clearly observed (Fig. 2a–d), which matched well with the widely accepted Zr-BTB structures[30,31]. By checking the stacking fashion between adjacent layers in the HAADF patterns carefully, we first found distinct Moiré patterns[32,33] across the ambient-dried 2-D Zr-BTB-FA samples (Fig. 2a–c and Supplementary Fig. 1), which was strong evidence to indicate skew and unaligned lattices between adjacent layers. Furthermore, we confirmed the three specific angles of 8°, 14°, and 30° between tilted nanosheets orientations by the comparison with our simplified $Zr_6$ clusters (Fig. 2a–c) and simulated supercell of two adjacent layers (Fig. 2i–k). The simulations in both models for three different angles were all well-matched with their Moiré patterns, respectively. Since the ordering of $Zr_6$ clusters was well-kept long the edge in the upper MOF nanosheet, we could even directly observe the twist angle of 14° between adjacent layers (Fig. 2b). The consistence of observed angles and simulated angles further confirmed the twisted phenomena happened within sub-nanometer between adjacent nanosheets. Considering the symmetry of clockwise and counterclockwise rotations between adjacent layers, as well as the symmetry of 2-D frameworks, the observation of above 8°, 14°, and 30° rotations actually spanned the random twisted stackings, indicating three representative conformations. It was worth noting that the random and extensive distribution of three main twisted angles (Supplementary Figs. 1 and 3) still could not rule out the possibilities of other twisted angles. At the same time, the three main conformations were still stable after 250 °C treatment possibly due to the disaligned hydrogen bonding and π–π stacking (Supplementary Fig. 5a–c).

This unique properties of adjacent layer stacking in Zr-BTB-FA nanosheets encouraged us to explore the tuning of stacking modes. After optimization, we found the soaking in ethanol together with heating at 80 °C for 6 h under vacuum was an efficient and simple way to untwist the tilted nanosheets stacking. The eclipsed conformation between adjacent layers was observed possibly because stronger interactions than the van der Waals forces were generated between each layers during the ethanol soaking and heating procedures. As a result, no Moiré patterns could be observed in all HAADF observations (Fig. 2d and Supplementary Figs. 2 and 4), firmly demonstrating the eclipsed conformation were stable structures. Considering the possible interactions between the adjacent nanosheets, we hypothesized covalent bonds of Zr–O–Zr between Zr-clusters formed during ethanol soaking and preheating processes[34] and induced the structure transformation. The fast Fourier transform (FFT) images obtained from the local regions on HAADF obviously

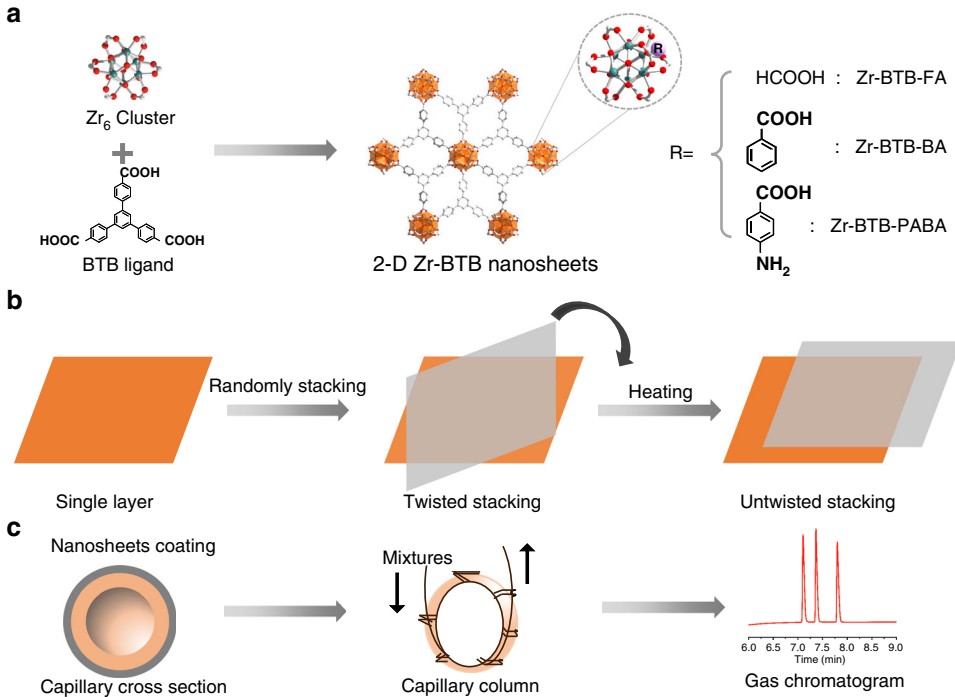

**Fig. 1** Functionalized Zr-BTB nanosheets with untwisted stacking mode for chromatographic separation. **a** Synthesis of 2-D Zr-BTB nanosheets with different functionalities via coordination assembling from $Zr_6$ clusters and BTB ligands. **b** Twisted mode between adjacent layers is obtained by randomly stacking, while untwisted mode is generated by heating the twisted nanosheets. **c** Fabrication of Zr-BTB nanosheets coated capillary column for the gas chromatographic separation of isomer mixtures

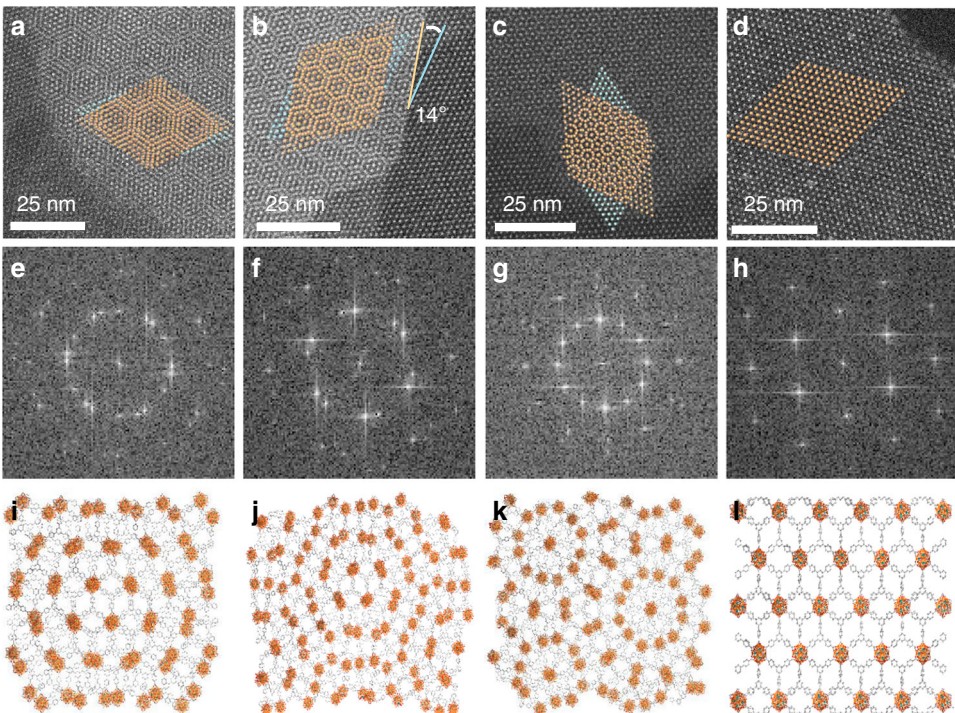

**Fig. 2** Morphology and Structural Simulation of 2-D twisted and untwisted Zr-BTB-FA nanosheets. HAADF images (**a–d**), FFT images (**e–h**) and simulated structures (**i–l**) of double layers with the rotation of 8° (**a**, **e**, **i**), double layers with the rotation of 14° (**b**, **f**, **j**) and double layers with the rotation of 30° (**c**, **g**, **k**), multilayers with the rotation of 0° (**d**, **h**, **l**)

also supported the rotations of adjacent layers and confirmed the untwisted feature of preheated Zr-BTB-FA nanosheets (Fig. 2e–h). These results revealed major structural difference between twisted and untwisted Zr-BTB-FA nanosheets.

**Characterization of twisted and untwisted Zr-BTB nanosheets.** To distinguish the different layers stacking between twisted and untwisted Zr-BTB-FA nanosheets, we rationalized structure by combining several pieces of information provided from PXRD, $N_2$ isothermal adsorption, AFM, DRIFTS, and molecular simulation. Further difference between two stacking modes of twisted and untwisted Zr-BTB-FA nanosheets were characterized with PXRD, which clearly showed eight reflection peaks, indexing to (0kl) reflections, such as the reflections of (011), (002), (022), (042), and (05-3) (Fig. 3a,b). The presence of only (0kl) reflections was strongly indicative of 2-D layered structures. It was worth mentioning that the main (011) and (002) planes of untwisted Zr-BTB-FA slightly shifted to lower diffraction angles compared to twisted Zr-BTB-FA, as shown in the Fig. 3a. This phenomenon manifested that the interplanar distances along the (011) and (002) direction in untwisted Zr-BTB-FA are larger than those in twisted Zr-BTB-FA. We speculated that this phenomenon contributed from the untwisted ordering process during heating.

To resolve the untwisted and twisted structures from the porous characteristics, both stacking modes were studied by $N_2$ sorption measurements and interpreted by BET theory. Significant higher adsorption properties were observed on Type-I isotherm of untwisted Zr-BTB-FA than that of twisted Zr-BTB-FA. The BET surface area of 228.8 $m^2 \cdot g^{-1}$ and 338.3 $m^2 \cdot g^{-1}$, as well as the total pore volumes of 0.076 $cm^3 \cdot g^{-1}$ and 0.115 $cm^3 \cdot g^{-1}$ for the twisted and untwisted Zr-BTB-FA nanosheets were obtained, respectively (Fig. 3c). In the horizontal view of the model, the interlayer distance of the untwisted nanosheets is shorter than the twisted ones (Supplementary Fig. 5). However, it was worth noting that the pores in vertical views of untwisted nanosheets are highly ordered and more accessible, while not all cavities between adjacent layers in the twisted stackings were effective, giving the different BET surface areas and pore volumes between untwisted and twisted nanosheets. The major pore sizes for twisted Zr-BTB-FA located at 7.4 and 11.7 Å, respectively, while untwisted Zr-BTB-FA located at 8.8 and 14.9 Å, respectively based on the nonlocal density functional theory (NLDFT) (Fig. 3d). The pores in twisted Zr-BTB-FA were observed to converge into one after heating, which could be seen from the side peak at 10 Å in untwisted sample. This convergence was well fitted to the simulation results. In the simulated structures, for each twisted nanosheets with angles of 8°, 14°, and 30°, the predictable pore sizes were found in two distinctive categories of 8.0 Å/10.6 Å, 7.6 Å/10.0 Å, and 8.0 Å/9.9 Å, respectively, while the predictable pore size for untwisted nanosheets was 9.0 Å (Supplementary Fig. 5). The experimental wide distribution of 11.7 Å in twisted Zr-BTB-FA was in accordance to the simulated sizes of larger pores in twisted nanosheets (Fig. 3d and Supplementary Fig. 5). After untwisted restacking process, the 8.8 Å pore contributed most to the total pore volumes, indicating highly ordered sub-nm cavities in the stacking, while there were still minor pore volumes contributed from pores of 14.9 Å. Thus, BET surface area, pore volume and pore size distribution are good indicators to characterize the untwisted and twisted structures.

The heating for untwisted stacking not only changed the stacking modes but also affected the nanosheet morphologies. Obvious thicker nanosheets stacking was obtained on TEM measurements comparing to the samples without heating (Supplementary Fig. 6). AFM revealed that the twisted Zr-BTB-FA stacks with the height of less than 8 nm (Fig. 3g and Supplementary Fig. 7), while untwisted Zr-BTB-FA demonstrated as much as 45 nm of thickness (Fig. 3h), indicating stronger interactions between adjacent layers.

Thermogravimetric analysis (TGA) plots and in-situ PXRD measurements evidenced both stacking modes of Zr-BTB-FA were thermally stable up to ~400 °C (Fig. 3e,f and Supplementary Fig. 8). The structural change from twisted to untwisted stacking was also revealed from TGA data. Although treated with heating at 80 °C for 6 h, the untwisted Zr-BTB-FA nanosheets lost as much as 8.5% from RT to 80 °C, mainly for readsorption of atmospheric water. From 80 °C to 400 °C, the untwisted Zr-BTB-FA counterintuitively gave as low as 3.5% of the gravimetric loss comparing with 6.8% for twisted Zr-BTB-FA. This phenomenon indicated the untwisted restacking process possibly involved not only the dehydration but also the removal of formate and even linkage between adjacent Zr clusters during the 6-h heating process at 80 °C.

To elucidate the interactions between adjacent layers during the transformation from twisted to untwisted stackings, H[1]NMR and DRIFTS of both samples were measured. Considering the coordination of $Zr_6$ cluster, the molar ratios of n($Zr_6$ cluster): n(HCOOH): n(BTB) = 1: 2: 2 for twisted Zr-BTB-FA, as well as 1: 0.5: 2 for untwisted Zr-BTB-FA were obtained in the H[1]NMR experiment (Supplementary Figs. 9 and 12). It indicated the 1.5 amount of formic acid loss on every $Zr_6$ cluster during the process of preheating, in agreement with our TGA data. This significant loss of formic acid under low temperature was probably due to the formation of ethoxy group in the ethanol mediated reactions, which recently were revealed by other groups[35]. In the DRIFTS, the ethoxy groups on $Zr_6$ clusters were obviously identified through the C–H bands at 2966 $cm^{-1}$, 2924 $cm^{-1}$, and 2865 $cm^{-1}$, as well as C–O bands at 1148 $cm^{-1}$, 1075 $cm^{-1}$, and 1044 $cm^{-1}$ (Supplementary Fig. 13), which were in accordance to other references[36]. The decrease of ethoxy groups in untwisted Zr-BTB-FA nanosheets were observed to indicate the further loss of ethoxy group during the heating procedures to form possibly the chemical linkages of Zr–O–Zr between Zr-clusters in the adjacent layers through the ethoxy group-mediated linkage reaction (Supplementary Fig. 14)[34]. These strong interactions from each Zr clusters between adjacent nanosheet layers created the untwisted stacking, which built the skeleton for highly ordered nanocavities. It was worth noting ethanol pretreatment is essential for the transformation from twisted to untwisted nanosheets in the heating/vacuum procedure. The heating of twisted nanosheets without ethanol at 250 °C still showed twisted structures (Supplementary Fig. 3), although this treatment gave 1.4 amount of formic acid loss on every $Zr_6$ cluster with the molar ratios of n($Zr_6$ cluster): n(HCOOH): n(BTB) = 1: 0.6: 2 (Supplementary Figs. 10 and 11). This control experiment demonstrated ethoxy group-mediated linkage reaction other than the loss of formic acid was the key to form untwisted structures.

To further tailor the chemical environment within the pores of nanosheets, we chose benzoic acid as modulator to form Zr-BTB-BA nanosheets and 4-aminobenzoic acid as post-synthetic linker to anchor onto the $Zr_6$ cluster of Zr-BTB-FA to result in Zr-BTB-PABA nanosheets (Supplementary Figs. 15–22). The installation of functional groups for 2-D Zr-BTB-BA and Zr-BTB-PABA were confirmed by H[1]NMR, PXRD, $N_2$ adsorption, and TEM measurements, respectively.

**Separation performance with *para*-selectivity.** The successful observation of sub-nanometer pores and decryption of the formation of untwisted stacking encouraged us to explore the separation performance due to the size matching between

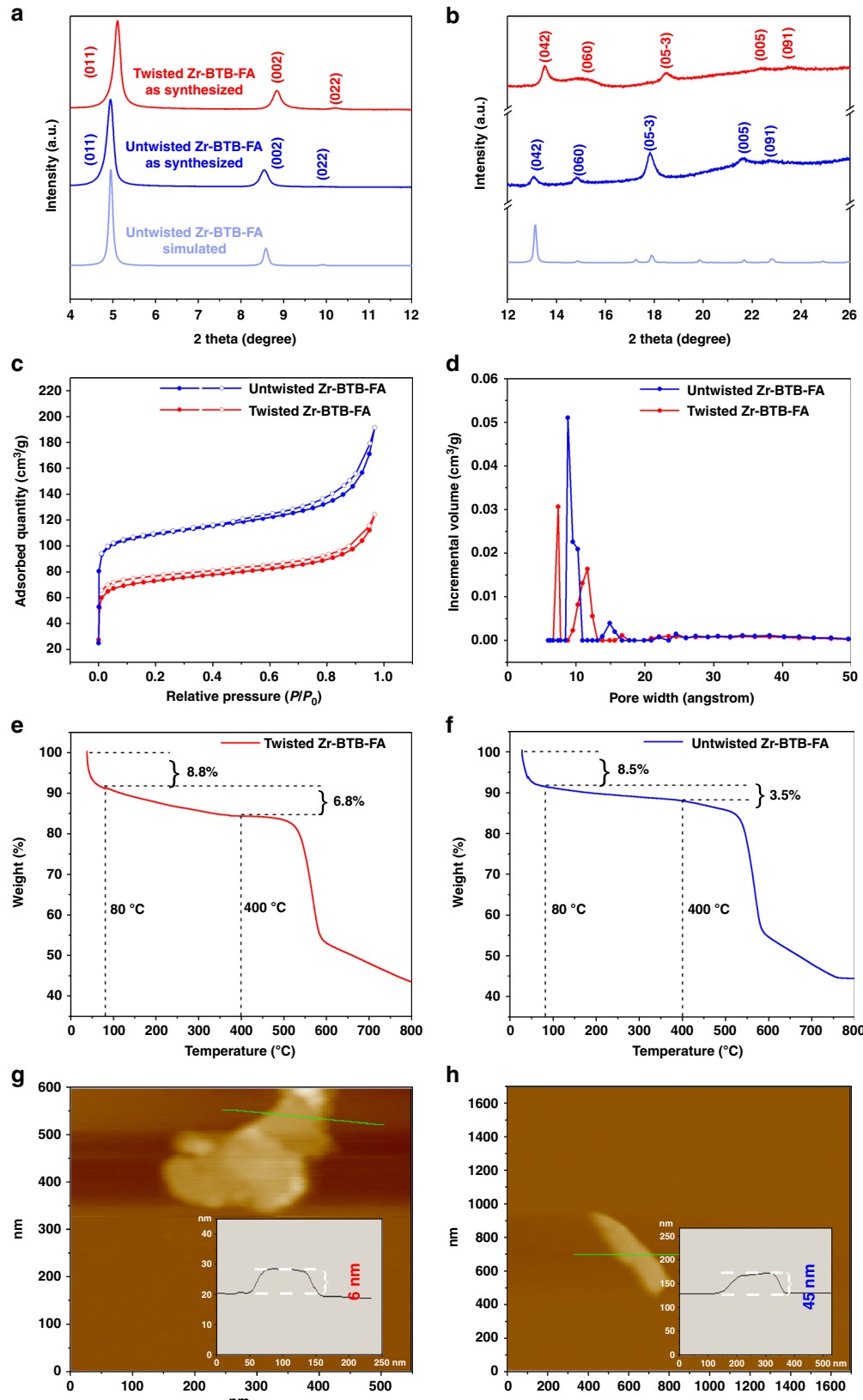

**Fig. 3** Characterization of 2-D Zr-BTB-FA nanosheets. **a**, **b** PXRD patterns of twisted and untwisted Zr-BTB-FA nanosheets compared with simulation of twisted structures from preferred orientation (100), which obtained at a scan rate of 1° min⁻¹. **c** $N_2$ adsorption-desorption isotherms of twisted and untwisted Zr-BTB-FA nanosheets. **d** Pore size distributions calculated via NLDFT method for twisted and untwisted Zr-BTB-FA nanosheets. **e**, **f** TGA curves of twisted and untwisted Zr-BTB-FA nanosheets, respectively. **g**, **h** AFM images of twisted and untwisted Zr-BTB-FA nanosheets and inset shows height profile along the green line, respectively

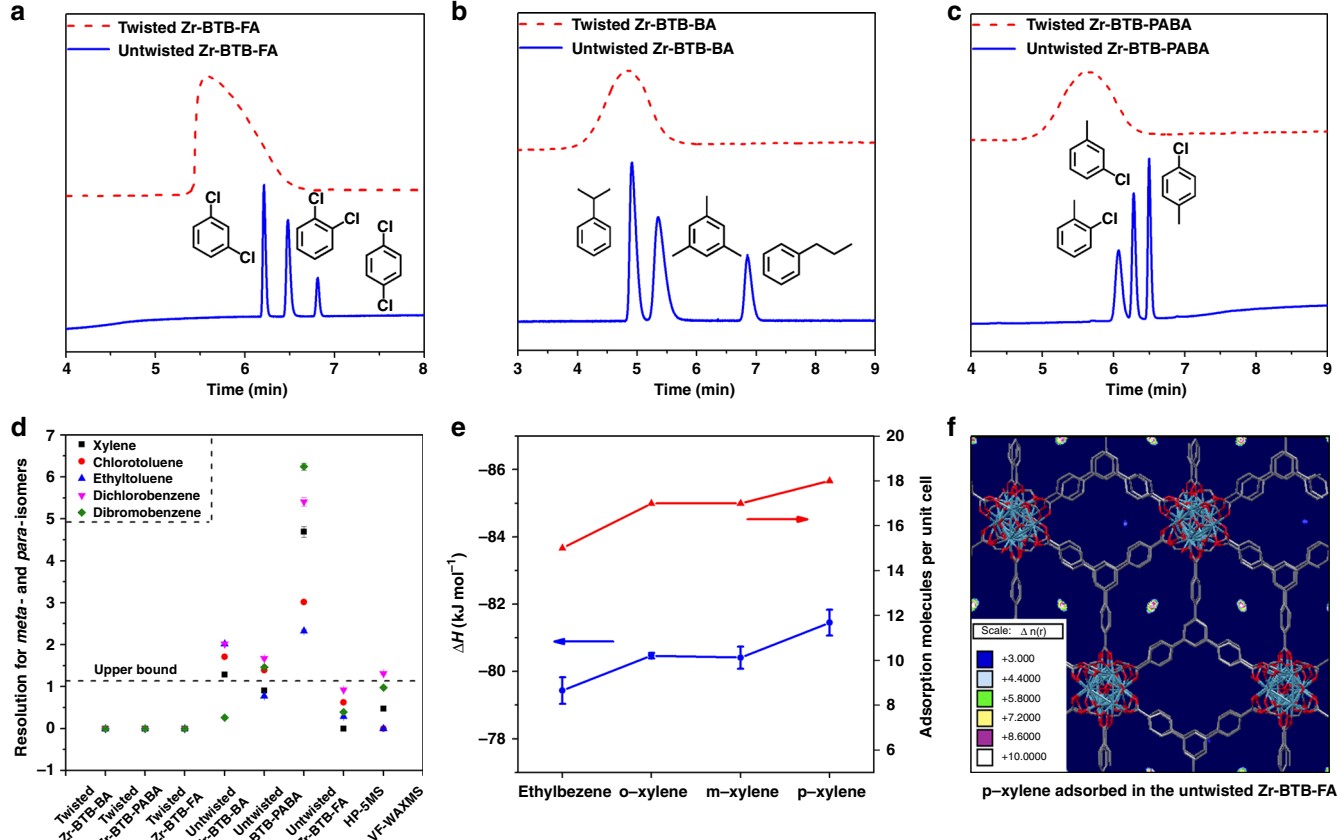

**Fig. 4** Separation performance and computational simulation of twisted and untwisted Zr-BTB nanosheets. **a** GC chromatogram on the Zr-BTB-FA coated column for the separation of dichlorobenzene isomers. **b** GC chromatogram on the Zr-BTB-BA coated column for the separation mixtures of n-propylbenzene, isopropylbenzene and 1,3,5-trimethylbenzene. **c** GC chromatogram on the Zr-BTB-PABA coated column for the separation of chlorotoluene isomers. **d** Resolution for *meta*-isomers and *para*-isomers on the twisted, untwisted Zr-BTB coated columns, HP-5MS and VF-WAXMS column. **e** Value of $\Delta H$ for ethylbenzene and xylene isomers on the untwisted Zr-BTB-FA coated column and corresponding adsorption capacities based on GCMC simulation. **f** Snapshots of density distribution for *p*-xylene adsorbed in the untwisted Zr-BTB-FA nanosheets at 523 K and $10^5$ Pa obtained from GCMC simulation

substitute benzene isomers and sub-nanocavities. The successful bound of 2-D Zr-BTB-FA nanosheets with twisted and untwisted different stacking models on the inner wall of capillary columns was supported by SEM images (Supplementary Figs. 23 and 24). To make a fair comparison, the thickness of twisted and untwisted nanosheets were both coated as 0.9 μm. It was worth mentioning that for twisted Zr-BTB-FA column, the restacking was largely prevented when heating in the column and the stacking modes did not change, because the well dispersion of thin nanosheets onto the inner wall to physically avoid the restacking. This was directly evidenced by the HAADF images of twisted and untwisted nanosheets after heating under the same coating temperature (Supplementary Figs. 3 and 4).

The untwisted Zr-BTB-FA nanosheets coated column showed significant separation performance with excellent selectivity for six groups of substituted aromatic isomers including ethylbenzene, xylene isomers, chlorotoluene isomers, ethyltoluene isomers, dichlorobenzene isomers, dibromobenzene isomers, n-propylbenzene, isopropylbenzene, and 1,3,5-trimethylbenzene (Fig. 4a and Supplementary Fig. 25). All selected structural isomers of benzene derivatives were well isolated from each other. At the same time, twisted Zr-BTB-FA almost had no separation ability for above isomers (Fig. 4a and Supplementary Fig. 26). Additional separations with linear alkanes and benzene homologs further confirmed the untwisted Zr-BTB-FA nanosheets column was superior than its twisted counterpart (Supplementary Figs. 27 and 28). The untwisted 2-D Zr-BTB-FA nanosheets coated

capillary column gave a theoretical plate number of 1980 plates·m⁻¹ with the practical thermal stability to at least 250 °C (Supplementary Figs. 29 and 30), which was confirmed by in-situ variable temperature PXRD with unchanged peaks of the (011) and (002) crystal planes (Supplementary Fig. 8). These phenomena identified the existence and separation capability of highly ordered sub-nanometer pores in the untwisted nanosheets, which outperformed its twisted counterpart.

It was noteworthy that all *para*-disubstituted aromatics (like p-xylene or p-chlorotoluene) always the last component out of isomers eluted from column (Supplementary Fig. 25), indicating the untwisted Zr-BTB-FA nanosheets had the stronger interaction with *para*-isomer. It was very different from the commercial column HP-5MS and other 3-D MOFs coated columns, like MOF-5[37], MIL-101 (Cr)[38], and UiO-66[39], which followed the order of boiling points. Since *para*-isomer is of industrial significance, such as *para*-xylene in polyester production, the separation of *para*-isomer from its isomer mixture is widely concerned. Our untwisted Zr-BTB-FA column demonstrated great potential towards this state-of-art application. For example, the separation resolution (R$_s$) between *para*-isomer and *meta*-isomer of xylene was enhanced from 0 on twisted Zr-BTB-FA column to 4.69 on the untwisted nanosheets column, which was superior to commercial capillary column with different polarity like HP-5MS and VF-WAXMS columns (Fig. 4d, Supplementary Figs. 31, 32 and Supplementary Table 1) and 3-D MOF columns, MIL-101 (Cr)[38] (R$_s$ = 1.25) and UiO-66[39] (R$_s$ = 1.00). In

addition, the relative standard deviation values were less than 0.12% for selectivity factor and less than 2.73% for $R_s$, respectively, on the untwisted Zr-BTB-FA for the separation of five isomer groups (Supplementary Table 1). Furthermore, identical separation performance was obtained in the separation of same isomers with 10-times injections (Supplementary Fig. 33), indicating its good column repeatability. The lifetime of the column was at least 16 months without the loss of separation efficiency (Supplementary Fig. 34), showing its feasibility for the potential applications. To verify the thermodynamic nature on the interactions between structural isomers and nanosheets, the enthalpy changes ($\Delta H$) and entropy changes ($\Delta S$) were calculated from the van't Hoff plots (Supplementary Figs. 35, 36 and Supplementary Table 2). The *para*-isomers had highest adsorption enthalpies, showing the higher energetic interactions with the nanosheets, giving a good explanation for the elution sequence of structural isomers from a thermodynamics view on the untwisted Zr-BTB-FA column.

To test the generality of the effects between twisted and untwisted nanosheets, we coated capillary columns with 2-D Zr-BTB-BA and Zr-BTB-PABA nanosheets of different stacking models. Expectably, untwisted Zr-BTB-PABA and Zr-BTB-BA nanosheets were much better than twisted counterparts with similar preference for *para*-isomer, indicating the untwisted restacking is a general methodology for MOF nanosheets with different functionalities (Fig. 4b,c and Supplementary Figs. 37–42 and Supplementary Tables 3, 4). Other control experiments, including 3-D Zr-BTB MOFs and Zr-BTB-FA micrometer-sized plates were also performed, showing no separation activity (Supplementary Figs. 43–45). These separation results illustrated a sharp contrast between twisted and untwisted MOF nanosheets due to their major structural and stacking difference, interpreting that sufficient and ordered stacked layers of nanosheets were prerequisites and played an important role in the high efficient separation.

Further elucidation on the structural and energetic interactions between isomers and nanosheets were investigated by computational simulation. The adsorption of the isomer molecules in the twisted and untwisted stacking modes of 2-D Zr-BTB-FA nanosheets were calculated by applying GCMC simulations, taking xylene isomers and ethylbenzene as examples. The adsorption simulation for each individual isomer was performed to study the retention of guest molecule and eliminate the effect from other isomers. The untwisted nanosheets with the simulated pore size of 9.0 Å was tested. The separation of four isomers was distinctly observed with the specific sequence in untwisted materials. The p-xylene showed adsorption capacities of 18 molecules per unit cell, which was the largest value among these four isomers (Fig. 4e,f and Supplementary Table 5). Ethylbenzene showed the lowest adsorption capacities of 15 molecules per unit cell. The adsorption capacities of oX and mX exhibited almost identical values. Such sequence was in good agreement with not only the chromatographic elution sequence (Supplementary Fig. 25a) but also the enthalpy changes, in which there were $81.45 \pm 0.27$ kJ mol$^{-1}$ and $79.43 \pm 0.28$ kJ mol$^{-1}$ for p-xylene and ethylbenzene (Supplementary Table 2). Furthermore, the examination of simulated snapshots of xylene isomers and ethylbenzene demonstrated that all four isomers were preferably adsorbed in the same location between the untwisted nanosheet stacking, indicating the observation of the highly selective nanocavities (Fig. 4f and Supplementary Fig. 46). On the other hand, the construction of the periodic twisted structures with specific angles (8°, 14°, and 30°) of 2-D Zr-BTB nanosheet stackings was not successful due to the failure of adding periodic boundary conditions to the simulated supercells. Thus, to elucidate the effect of pore size, we further compared the stacking structures with larger pore size. The material with pore size as large as 10.35 Å, showed no separation effect for four isomers, corresponding to the same value

molecules per unit cell, indicating that unique pore size formed in the untwisted structure was the key factor in the highly selective isomer separation (Supplementary Table 6).

In summary, we discovered three main conformations for the random stacking layers of Zr-BTB-FA MOF nanosheets with twisted angles of 8°, 14°, and 30°. The proposed preheating strategy was efficient to untwist the stacking and create ordered sub-nanometer pores by the possible ethanol-assistant cluster linkage reaction. The confirmed nanopores in the nanosheets were capable for the high resolution separation of six groups of isomers. The *para*-isomer selectivity of untwisted nanosheet was highly practical and supported by GCMC simulation. The investigation of twisted and untwisted stacking of MOF nanosheets may open new opportunity to control the interactions within sub-nanometer distance for the design of functional porous solids.

## Methods

**Synthesis of 2-D Zr-BTB-FA nanosheets**. 2-D Zr-BTB nanosheets with formic acid as modulator (Zr-BTB-FA nanosheets) were synthesized, briefly, H$_3$BTB (12.50 mg) and ZrCl$_4$ (10.12 mg) were dissolved in 5 mL DMF in a 22 mL pyrex vial under ultrasonication for 10 min. Formic acid (1.11 g) and water (60 μL) were then added to the clear solution. The vial was sealed and placed in an oven at 120 °C for 48 h, then cooled to room temperature. The white solids were obtained by centrifugation, then washed with DMF and ethanol sequentially to remove unreacted reagents in the pore. To avoid the restacking between nanosheets, the as-synthesized products were kept in ethanol solution, and marked as twisted Zr-BTB-FA nanosheets.

**Synthesis of restacked untwisted 2-D Zr-BTB-FA nanosheets**. The as-synthesized 2-D Zr-BTB-FA nanosheets were heated under vacuum at 80 °C for 6 h to obtain 2-D Zr-BTB-FA nanosheets with restacked untwisted structures. These untwisted structures were confirmed by many characterization techniques and molecular simulation. To distinguish with twisted Zr-BTB-FA nanosheets, restacked sample was marked as untwisted Zr-BTB-FA nanosheets.

**Computational simulation**. GCMC simulations were carried out using the developed CADSS software to calculate the adsorption capacities of xylene isomers and ethylbenzene in 2-D Zr-BTB-FA nanosheets using the rationalized structure. Our calculations were performed at 523 K and 10$^5$ Pa considering that the conditions of separation performance of this MOF nanosheet coated capillary column in the experiment. Single component adsorption capacities were simulated at least 5,000,000 equilibrium steps, followed by 5,000,000 production steps. Each movement of xylene isomers and ethylbenzene molecules consisted energy-biased insertion and deletion, rotation and translation. For the force field, Lennard–Jones parameters were taken from the DREIDING force field[40] for the atoms in framework, except for zirconium (taken from UFF force field[41]), which is absent in DREIDING force field. For xylene isomers and ethylbenzene, the molecules were treated as rigid ones and all atoms were defined explicitly except for the methyl groups that were considered as single spheres. The L-J parameters were obtained from the TraPPE-Extension force field[42], which has been proved to be able to well reproduce the experiment adsorption data of xylene in Zr-MOF, such as UiO-66(Zr).

## Data availability

The data that support the findings of this study are either provided in the Article and its supplementary information or are available from the authors upon reasonable request.

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

## Acknowledgements

This work is financially supported from the NSFC (No. 21505076), the Young Elite Scholar Support (YESS) Program from CAST (YESS20150009), the Program of Jiangsu Specially Appointed Professor, the NSF of Jiangsu Province of China, the Innovation Team Program of Jiangsu Province of China, and the Priority Academic Program Development of Jiangsu Higher Education Institutions. We acknowledge Prof. Zhen-Jie Zhang and Qi Yu for the DRIFTS measurements. We gratefully acknowledge Dr. Kui Tan for the help and insightful discussion. We thankfully acknowledge Prof. Chongli Zhong and Prof. Qingyuan Yang (Beijing University of Chemical Technology) for providing their in-house code CADSS (Complex Adsorption and Diffusion Simulation Suite) in GCMC simulations.

## Author contributions

Z.-Y.G. conceived the idea and supervised the research. Z.-R.T. performed the separation and characterization experiments and analyzed the data. J.-X.W. synthesized the nanosheets samples. M.X. draw the figures and performed the SEM measurements. W.-Q.T. performed GC measurements. Q.-H.Z. and L.G. conducted the HRTEM measurements and analyzed the data. D.-H.L. and Y.-J.Z. carried out the GCMC simulation. Z.-Y. G. and Z.-R.T. discussed the experimental data and wrote the paper.

## Additional information

**Competing interests:** The authors declare no competing interests.

