## [Peer Review File · Nature Communications]

Reviewers' Comments

Reviewer #1 (Remarks to the Author):

There is some interesting phenomena revealed. The main issue is that column separation is not as important as bulky material separation and also much easier. Column separation of xylene mixture has been established before, so this work is weak and not suitable for the publication in Nature Communications.

Reviewer #2 (Remarks to the Author):

The authors observed an interesting stacking of two-dimensional (2-D) metal-organic framework (MOF) nanosheets, which could greatly influence the adsorption capacity and selectivity over several series of aromatic isomers. But in this case, the paper does not provide significant enough data sets to warrant publication in the current form. The following specific comments are provided for feedback to the author.

1. Do the claimed three main conformations of stacking include all the possibilities or just a random observation? This conclusion should be clarified. Why these modes of stacking are stable in 2-D twisted Zr-BTB-FA nanosheets?
2. Considering the substantial weight loss after heating under vacuum, it is believed that the spacing molecules like FA were removed. The twisted structure might stably exist only in the presence of spacing molecule like FA. So what is the real specific role of FA(BA, PABA) in the crystalline structure? This important question remains unsolved.
3. What is the predictable micropore size from as-synthesized crystalline structure and why had it been changed (such as from 7.4 Å to 8.8 Å)? Besides, other relevant data of pore or surface properties should be provided for comparison, such as BET surface area, micropore volume, and so on. From the N₂ adsorption/desorption curve, a big change in BET surface area happened after the structure change. Is there any possibility that removal of blocked FA molecule in twisted 2-D Zr-BTB could explain the difference of capacity and selectivity between twisted and untwisted samples?
4. The pores in twisted Zr-BTB-FA located at 7.4 and 11.7 Å seem to converge into one after heating, which can be seen from the side peak at about 10 Å in untwisted sample. The authors should further clarify what did these two peaks represent and the reasons of this change?
5. Why use the HP-5MS column for selectivity comparison? Since it is well-known that the nonpolar HP-5MS column could barely separate PX and MX, these comparisons seem to be unconvincing. The authors also claimed "The thickness of twisted and untwisted nanosheets were approximately 0.2 and 0.9 μm, respectively", which might be unreasonable to compare the selectivities by different coating thickness. This problem seriously impedes the significance of present paper.
6. The authors claimed "It was worth mentioning that for twisted Zr-BTB-FA column the restacking was largely prevented when heating in the column because the well dispersion of thin nanosheets onto the inner wall to physically avoid the restacking." , which needs direct proof.

7. Some parts of the paper are hard to digest and even wrong, such as the last two paragraph before conclusion. "With larger spacing length of 10.35 Å, although untwisted, the material showed identical adsorption capacities for four isomers, corresponding to the same value molecules per unit cell, indicating the failure to the isomer separation (Supplementary Table 4)." What is the origin of 10.35 Å? Why the simulation of untwisted material indicated the failure to the isomer separation but the experiment showed different? In section 6 of supplement info, "Formic acid (1.910 g)" should be "formic acid". Manuscript will benefit from the careful proof reading by the native speaker

Reviewer #3 (Remarks to the Author):

The topic considered in this communication is important, and the results look interesting; however, it is unclear if they justify publication as a Nature Communication. Response: We highly appreciate the referee for the supportive comments. We have addressed all the concerns from Reviewer #3, especially in the comparison of separation performance with commercial HP-5MS and VF-WAXMS column and elucidation of transformation from twisted to untwisted nanosheets.

1. I agree that it is impressive as the authors note claim in second page of the introduction that this is the first fabrication of MOF nanosheets-coated gas chromatographic (GC) capillary columns. This fact notwithstanding, despite the greatly enhanced separation capacity for the untwisted stacked nanosheets, relative to the twisted nanosheets, it is NOT clear to this reviewer that the results are (as they claim is shown in Fig. 1c) also much better even than the commercial column for para/meta isomer separation.
2. In fact, while the separation efficiency of the untwisted, restacked nanosheet in Fig.4 and in supplemental material Fig. 18 and 27 are good, it is unclear that the separation efficiency is statistically better than the commercially available gas chromatograph (GC) capillary column results. If the authors would clarify why they say that the untwisted restacked nanosheets are truly better than the current capillary column performance for the various cases shown, it would be more impressive.
3. Even without such proof, it still may be justifiable to publish the results, since the work does represent a novel application of MOF's, but but overstatement is not appropriate, and unless I am missing something they are overstating the situation.
4. I am also a bit worried that the performance of untwisted stacked nanosheets will not be stable, since as noted there are at least three thermodynamic-favored (not "flavored" as they say) conformations. They note three specific angles of 8, 14 and 30 between tilted nanosheets stacking of Zr-BTB-FA were observed. While they note untwisted stacking was achieved through a preheating strategy to get thermodynamic-favored and highly ordered nanosheets, it was not clear to this referee why the three specific angular plate configurations existed per se.
5. The authors hypothesize that their "preheating strategy" (80 oC for 6 h under vacuum) led to the desirable (and hopefully stable) untwisted conformations, since they believe the eclipsed conformation between adjacent layers were observed because the van der Waals interactions between each layers were too weak to prohibit the rotations in the

heating-driven restacking procedures. They suggest the eclipsed structure was thermodynamically favored due to interactions (which were not clearly defined) that were stronger than van der Waals forces. It is not clear to this reviewer what the claimed interactions were or why the resultant conformations would remain stable for an extended period. The authors note that they exposed the untwisted 2-D Zr-BTB-FA nanosheets coated capillary column to 250 oC (Supplementary Fig. 22 and 23). It was not clear to this reviewer if they showed the SAME PERFORMANCE before and after this exposure. If they did this explicit check and would include a figure like that in Fig. 4 after the exposure, I would agree that the work is worthy of publication.

6. Although it is a cosmetic issue, there are many typos and English issues in the manuscript, and these should be remedied by having a native English speaker read the paper.

Response to Reviewers' Comments

Reviewer #1

Comments:

There is some interesting phenomena revealed. The main issue is that column separation is not as important as bulky material separation and also much easier. Column separation of xylene mixture has been established before, so this work is weak and not suitable for the publication in Nature Communications.

Response: Thank you for your comments. There are different types of chromatographic column separation. The specific column separation, like simulated moving bed (SMB) chromatography is of great importance and widely used in the industrial separation of pharmaceutical intermediates and xylene isomers. The novelty of this study is the discovery of the unique separation pores during the stacking between adjacent untwisted MOF nanosheets. Our research is mainly focused on the twisted/untwisted MOF nanosheets and their distinctive isomer separation mechanism, which has never been explored before. Six groups of substituted benzene isomers are employed as model isomers to test the separation. This discovery will provide a new synthetic route in the rational design of desirable pore structures in 2-D MOF nanosheets and their separation applications.

Reviewer #2

Comments:

The authors observed an interesting stacking of two-dimensional (2-D) metal-organic framework (MOF) nanosheets, which could greatly influence the adsorption capacity and selectivity over several series of aromatic isomers. But in this case, the paper does not provide significant enough data sets to warrant publication in the current form. The following specific comments are provided for feedback to the author.

Response: We highly appreciate the referee for the supportive comments. We have addressed all the concerns from Reviewer #2. The manuscript is significantly strengthened, especially in the characterization and elucidation of chemical environments of micropores. Due to the nature of 2-D MOF nanosheets, their crystalline structures, including the conformations of whole nanosheets and the acidic modulators, could not be directly solved by the single crystal diffraction techniques. Thus, we combined modeling and high-angle annular dark field (HAADF) images to characterize the nanosheets and elucidate the reason of stability and role of modulators.

1. Do the claimed three main conformations of stacking include all the possibilities or just a random observation? This conclusion should be clarified. Why these modes of stacking are stable in 2-D twisted Zr-BTB-FA nanosheets?

Response:

Response: We highly appreciate the referee for the constructive comments. We have performed new HRTEM experiments and collected more than 100 HAADF images for twisted 2-D MOF nanosheet samples. Moiré patterns were extensively observed in most of the images. More representative images were added and analyzed carefully. We found the three main conformations with twisted angles of 8°, 14° and 30° were extensively and randomly distributed across the sample. At the same time, we could not rule out the possibilities of other twisted angles. “It was worth noting that the random and extensive distribution of three main twisted angles could not rule out the possibilities of other twisted angles (Supplementary Fig. 1 and 3)” has now been added. (Please see Page 2, right column, 1st paragraph in the main article). “thermodynamic favored conformation” has now been changed to “representative conformations” (Please see Page 1, abstract; Page 2, right column, 1st paragraph in the main article).

The stability of three main conformations in the twisted sample was further checked under heating at 250 °C in the ambient condition. The Moiré patterns were well kept, indicating that the three main conformations are stable in this condition. At the same time, we applied modelling to get supercell of two adjacent nanolayers with different angles. The simulations at molecular level for different angles were well-matched with their Moiré patterns, respectively. The optimized structures indicated that they are stable possibly due to the disaligned hydrogen bonding and π - π stacking. “The simulations in both models for three different angles were all well-matched with their Moiré patterns, respectively. At the same time, the three main conformations were still stable after 250 °C treatment possibly due to the disaligned hydrogen bonding and

π - π stacking (Supplementary Fig.5a-c).” has now been added. (Please see Page 2, right column, 1st paragraph in the main article).

Supplementary Figure 3. HAADF images of 2-D twisted Zr-BTB-FA nanosheets with Moiré patterns. (a, b and c) twisted Zr-BTB-FA nanosheets with the rotation of 8° , (d and e) twisted Zr-BTB-FA nanosheets with the rotation of 14° , (f, g, h and i) twisted Zr-BTB-FA nanosheets with the rotation of 30° . In order to simulate environment in the capillary, the sample was paved on a watch glass and placed in the GC oven to experienced temperature program for 250°C before HRTEM measurements.

Fig. 2i-l simulated structures (i-l) of double layers with the rotation of 8° (i), double layers with the rotation of 14° (j) and double layers with the rotation of 30° (k), multilayers with the rotation of 0° (l).

Supplementary Figure 5. The predictable pore size of 2-D simulated Zr-BTB-FA structure and the interaction between layers. (a) twisted Zr-BTB-FA with the rotation of 8° , (b) twisted Zr-BTB-FA with the rotation of 14° , (c) twisted Zr-BTB-FA with the rotation of 30° , (d) untwisted Zr-BTB-FA with the rotation of 0° .

2. Considering the substantial weight loss after heating under vacuum, it is believed that the spacing molecules like FA were removed. The twisted structure might stably exist only in the presence of spacing molecule like FA. So what is the real specific role of FA(BA, PABA) in the crystalline structure? This important question remains unsolved.

Response: Thank you very much for pointing this out. We totally agree with

Reviewer #2 that FA molecules were removed after heating under vacuum in the transformation from twisted nanosheets to untwisted nanosheets. Furthermore, from the ^1H -NMR data, we could quantify FA in both samples. “Considering the coordination of Zr_6 cluster, the molar ratios of $n(\text{Zr}_6 \text{ cluster}): n(\text{HCOOH}): n(\text{BTB})=1: 2: 2$ for twisted Zr-BTB-FA as well as $1: 0.5: 2$ for untwisted Zr-BTB-FA were obtained in the ^1H -NMR experiment (Supplementary Fig. 9). It indicated the 1.5 amount of formic acid loss on every Zr_6 cluster during the process of preheating, in agreement with our TGA data. This significant loss of formic acid under low temperature was probably due to the formation of ethoxy group in the ethanol mediated reactions, which recently were revealed by other groups.” has now been added. (Please see Page 5, left column, 4th paragraph in the main article).

We performed control experiment to elucidate the key factor in the transformation from twisted to untwisted nanosheets. The heating of twisted structure without ethanol at 250 °C still gave twisted structures but with less FA. “It was worth noting ethanol pretreatment is essential for the transformation from twisted to untwisted nanosheets in the heating/vacuum procedure. The heating of twisted nanosheets without ethanol at 250 °C still showed twisted structures (Supplementary Fig.3), although this treatment gave 1.4 amount of formic acid loss on every Zr_6 cluster with the molar ratios of $n(\text{Zr}_6 \text{ cluster}): n(\text{HCOOH}): n(\text{BTB})=1: 0.6: 2$ (Supplementary Fig.10). This control experiment demonstrated ethoxy group-mediated linkage reaction other than the loss of formic acid was the key to form untwisted structures.” has now been added. (Please see Page 5, right column, 1st paragraph in the main article).

The roles of FA in the formation of twisted structures and the transformation from twisted to untwisted structures still need further exploration. Based on the current data, we could summarize the overall picture that specific amount of FA is the important factor in the formation of twisted nanosheets. The asymmetrical arrangements of two FA molecules in one Zr_6 clusters could possibly provide driving forces to form the twisted stacking with different angles (Please see Page 16, Supplementary Fig. 12 in the Supplementary Information). However, after the formation of twisted structure, FA could be removed to obtain the twisted nanosheets or to form untwisted nanosheets totally depending on the absence and existence of ethanol pretreatment (Please see Supplementary Fig. 14 in the Supplementary Information).

3. What is the predictable micropore size from as-synthesized crystalline structure and why had it been changed (such as from 7.4 Å to 8.8 Å)? Besides, other relevant data of pore or surface properties should be provided for comparison, such as BET surface area, micropore volume, and so on. From the N_2 adsorption/desorption curve, a big change in BET surface area happened after the structure change. Is there any possibility that removal of blocked FA molecule in twisted 2-D Zr-BTB could explain the difference of capacity and selectivity between twisted and untwisted samples?

Response: Thank you very much for pointing this out. From the modeling of twisted and untwisted Zr-BTB-FA nanosheets, we have obtained the predictable micropore sizes. These predictable pore sizes are in accordance to the observed experimental pore size distribution and explained well the convergence from two

groups of pores in twisted nanosheets to one in the untwisted nanosheets. “The pores in twisted Zr-BTB-FA were observed to converge into one after heating, which could be seen from the side peak at 10 Å in untwisted sample. This convergence was well fitted to the simulation results. In the simulated structures, for each twisted nanosheets with angles of 8°, 14°, and 30°, the predictable pore sizes were found in two distinctive categories of 8.0 Å/10.6 Å, 7.6 Å/10.0 Å, and 8.0 Å/9.9 Å, respectively, while the predictable pore size for untwisted nanosheets was 9.0 Å (Supplementary Fig.5).....Thus, BET surface area, pore volume and pore size distribution are good indicators to characterize the untwisted and twisted structures.” has now been added. (Please see Page 5, left column, 1st paragraph in the main article).

“The BET surface area of 228.8 m²·g⁻¹ and 338.3 m²·g⁻¹ as well as the total pore volumes of 0.076 cm³·g⁻¹ and 0.115 cm³·g⁻¹ for the twisted and untwisted Zr-BTB-FA nanosheets were obtained, respectively (Fig. 3c). It was worth noting that the pores in untwisted nanosheets are highly ordered and more accessible, while not all cavities between adjacent layers in the twisted stackings were effective, giving the different BET surface areas and pore volumes.” has now been added. (Please see Page 3, right column, 2nd paragraph in the main article). Also, the corresponding data of untwisted Zr-BTB-BA and Zr-BTB-PABA nanosheets were added in the supplementary information. (Please see Page 26, Supplementary Fig. 21 in the Supplementary Information).

The activation of both twisted and untwisted samples included the washing with DMF and ethanol and pre-heating at 120 °C under vacuum overnight before the N₂ adsorption measurement. This process eliminated the possible free FA in the pores. To further confirm it, we measured the final washing solvent with NMR. It gave no FA peaks. (Please see Page 15, Supplementary Fig. 11 in the Supplementary Information).

Supplementary Figure 11. ^1H NMR of 2-D (a) $2.6 \text{ mmol}\cdot\text{L}^{-1}$ HCOOH and (b) the last ethanol washing eluent for twisted Zr-BTB-FA nanosheets. There is no HCOOH in the washing eluent, indicating no trapped FA in the Zr-BTB nanosheets.

4. The pores in twisted Zr-BTB-FA located at 7.4 and 11.7\AA seem to converge into one after heating, which can be seen from the side peak at about 10\AA in untwisted sample. The authors should further clarify what did these two peaks represent and the reasons of this change?

Response: We thank the referee for pointing this out. From the modeling of twisted and untwisted Zr-BTB-FA nanosheets, we have obtained the predictable micropore sizes. These predictable pore sizes are in accordance to the observed experimental pore size distribution and explained well the convergence from two groups of pores in twisted nanosheets to one in the untwisted nanosheets. “The pores in twisted Zr-BTB-FA were observed to converge into one after heating, which could be seen from the side peak at 10\AA in untwisted sample. This convergence was well fitted to the simulation results. In the simulated structures, for each twisted nanosheets with angles of 8° , 14° , and 30° , the predictable pore sizes were found in two distinctive categories of $8.0\text{\AA}/10.6\text{\AA}$, $7.6\text{\AA}/10.0\text{\AA}$, and $8.0\text{\AA}/9.9\text{\AA}$, respectively, while the predictable pore size for untwisted nanosheets was 9.0\AA (Supplementary Fig.5)..... Thus, BET surface area, pore volume and pore size distribution are good indicators to characterize the untwisted

and twisted structures.” has now been added. (Please see Page 5, left column, 1st paragraph in the main article).

5. Why use the HP-5MS column for selectivity comparison? Since it is well-known that the nonpolar HP-5MS column could barely separate PX and MX, these comparisons seem to be unconvincing. The authors also claimed "The thickness of twisted and untwisted nanosheets were approximately 0.2 and 0.9 μm , respectively", which might be unreasonable to compare the selectivities by different coating thickness. This problem seriously impedes the significance of present paper.

Response: Thank you for your suggestion. Considering that HP-5MS is nonpolar column, we now add comparison with the commercial polyethylene glycol VF-WAXMS polar column. The new separation result demonstrated that some of *m/p*-xylene and *m/p*-chlorotoluene were not separated on VF-WAXMS column. Moreover, our untwisted Zr-BTB-FA coated capillary column gave larger separation selectivity factor and better resolution of *meta*- and *para*- isomers than commercial HP-5MS and VF-WAXMS columns. “which was superior to commercial capillary column with different polarity like HP-5MS and VF-WAXMS columns (Fig. 4d, Supplementary Fig. 31,32 and Supplementary Table 1).” has now been added. (Please see Page 6, left column, 2nd paragraph in the main article).

To make a fair comparison, the new column with coating thickness of 0.9 μm was prepared for the twisted Zr-BTB-FA and tested for the separation performance. The column showed no separation for substituted benzene isomers. All the related chromatograms were replaced with the new column. “To make a fair comparison, the thickness of twisted and untwisted nanosheets were both coated as 0.9 μm .” has now been added. (Please see Page 5, right column, 3rd paragraph in the main article).

Fig. 4d Resolution for *meta*- and *para*-isomers on the twisted, untwisted Zr-BTB coated columns, HP-5MS and VF-WAXMS column.

6. The authors claimed "It was worth mentioning that for twisted Zr-BTB-FA column the restacking was largely prevented when heating in the column because the well dispersion of thin nanosheets onto the inner wall to physically avoid the restacking.", which needs direct proof.

Response: Thank you for your constructive suggestion. In light of your suggestion, the HAADF images were added to investigate nanosheets stacking modes after the heating process in the capillary column. In order to simulate the environment, we paved twisted and untwisted Zr-BTB-FA nanosheets on the watch glass, respectively, then placed them in the GC oven to exposed nanosheets to 250 °C for the same temperature program, after that we scraped nanosheets from glass for HRTEM measurements. The HAADF images indicated the sample with twisted Zr-BTB-FA nanosheets still had obvious Moiré patterns, and untwisted sample still had no Moiré patterns. Therefore, the heating after coating twisted and untwisted nanosheets in the capillary columns did not changed their stacking modes "the stacking modes did not change...This was directly evidenced by the HAADF images of twisted and untwisted nanosheets after heating under the same coating temperature" has now been added. (Please see Page 5, right column, 3rd paragraph in the main article).

7. Some parts of the paper are hard to digest and even wrong, such as the last two paragraph before conclusion. "With larger spacing length of 10.35 Å, although untwisted, the material showed identical adsorption capacities for four isomers, corresponding to the same value molecules per unit cell, indicating the failure to the isomer separation (Supplementary Table 4)." What is the origin of 10.35 Å? Why the simulation of untwisted material indicated the failure to the isomer separation but the experiment showed different? In section 6 of supplement info, "Formic acid (1.910 g)" should be "formic acid". Manuscript will benefit from the careful proof reading by the native speaker.

Response: We thank the referee for pointing this out. We have performed the simulation for twisted nanosheets to get the supercells for each angle. However, the construction of the periodic twisted structures with three angles (8°, 14° and 30°) of nanosheet stackings was not successful due to the failure of adding periodic boundary conditions to the supercells. To perform a fair simulation comparison and elucidate the pore size effect, we built a new structure with larger pore size under the untwisted nanosheets model and found the pore size is the key factor to affect the separation results.

We have now rewritten the paragraph and changed the interlayer spacing length to pore size for the better logic flow. "On the other hand, the construction of the periodic twisted structures with specific angles (8°, 14° and 30°) of 2-D Zr-BTB nanosheet stackings was not successful due to the failure of adding periodic boundary conditions to the simulated supercells. Thus, to elucidate the effect of pore size, we further compared the stacking structures with larger pore size. The material with pore size as large as 10.35 Å, showed no separation effect for four isomers, corresponding to the same value molecules per unit cell, indicating that pore size was the key factor in the

highly selective isomer separation (Supplementary Table 6).” has now been added. (Please see Page 7, left column, 1st paragraph line in the main article).

Thank you very much for pointing this out. We have corrected the word of "formic acid", "van der Waals interactions", "angle". (Please see Page 7, right column, 2nd paragraph; Page 3, left column, 1st paragraph; Page 2, right column 1st paragraph in the main article).

Reviewer #3

Comments:

The topic considered in this communication is important, and the results look interesting; however, it is unclear if they justify publication as a Nature Communication.

Response: We highly appreciate the referee for the supportive comments. We have addressed all the concerns from Reviewer #3, especially in the comparison of separation performance with commercial HP-5MS and VF-WAXMS column and elucidation of transformation from twisted to untwisted nanosheets.

1. I agree that it is impressive as the authors note claim in second page of the introduction that this is the first fabrication of MOF nanosheets-coated gas chromatographic (GC) capillary columns. This fact notwithstanding, despite the greatly enhanced separation capacity for the untwisted stacked nanosheets, relative to the twisted nanosheets, it is NOT clear to this reviewer that the results are (as they claim is shown in Fig. 1c) also much better even than the commercial column for para/meta isomer separation.

Response: Thank you for your positive comments. Considering that HP-5MS is nonpolar column, we now add comparison with the commercial polyethylene glycol VF-WAXMS polar column. The new separation result demonstrated that some of m/p-xylene and m/p-chlorotoluene were not separated on VF-WAXMS column. Moreover, our untwisted Zr-BTB-FA coated capillary column gave larger separation selectivity factor and better resolution of *meta*- and *para*- isomers than commercial HP-5MS and VF-WAXMS columns. “which was superior to commercial capillary column with different polarity like HP-5MS and VF-WAXms columns (Fig. 4d, Supplementary 31,32 and Supplementary Table 1).” has now been added. (Please see Page 6, left column, 2nd paragraph in the main article).

2. In fact, while the separation efficiency of the untwisted, restacked nanosheet in Fig.4 and in supplemental material Fig. 18 and 27 are good, it is unclear that the separation efficiency is statistically better than the commercially available gas chromatograph (GC) capillary column results. If the authors would clarify why they say that the untwisted restacked nanosheets are truly better than the current capillary column performance for the various cases shown, it would be more impressive.

Response: We highly appreciate the referee for constructive comments. The repeatability of untwisted Zr-BTB-FA, HP-5MS and VF-WAXms column were performed by separation of five groups of substituted benzene isomers, and were evaluated by relative standard deviation (RSD%) values on their separation selectivity factor and resolution (n=7). “In addition, the relative standard deviation values were less than 0.12% for selectivity factor and less than 2.73% for R_s , respectively, on the untwisted Zr-BTB-FA for the separation of five isomer groups (Supplementary Table 1)” has now been added. (Please see Page 6, left column, 2nd paragraph and Fig. 4d in the main article). Moreover, other data for Zr-BTB-BA and Zr-BTB-PABA nanosheets

coated columns were also updated. (Please see Page 49, Supplementary Table 3 in the Supplementary Information).

Fig. 4d Resolution for *meta*- and *para*-isomers on the twisted, untwisted Zr-BTB coated columns, HP-5MS and VF-WAXMS column.

3. *Even without such proof, it still may be justifiable to publish the results, since the work does represent a novel application of MOF's, but but overstatement is not appropriate, and unless I am missing something they are overstating the situation.*

Response: Thank you for your kind suggestion! To eliminate the overstatements, we have now changed “commercial column” to “commercial HP-5MS and VF-WAXMS columns” for exact expression. (Please see Page 1, abstract; Page 2, left column 1st paragraph; Page 6, left column, 2nd paragraph; in the main article).

4. *I am also a bit worried that the performance of untwisted stacked nanosheets will not be stable, since as noted there are at least three thermodynamic-favored (not “flavored” as they say) conformations. They note three specific angles of 8, 14 and 30 between tilted nanosheets stacking of Zr-BTB-FA were observed. While they note untwisted stacking was achieved through a preheating strategy to get thermodynamic-favored and highly ordered nanosheets, it was not clear to this referee why the three specific angular plate configurations existed per se.*

Response: Good question! The stability of separation performance for untwisted Zr-BTB-FA and the transformation of twisted to untwisted nanosheets have been checked.

First, the stability and stacking mode of untwisted Zr-BTB-FA was further investigated when exposed nanosheets to 250 °C. After that we scraped the sample for HRTEM measurements. The untwisted Zr-BTB-FA still have no Moiré patterns,

indicating untwisted Zr-BTB-FA are stable in this condition. “the stacking modes did not change...This was directly evidenced by the HAADF images of twisted and untwisted nanosheets after heating under the same coating temperature” has now been added. (Please see Page 5, right column, 3rd paragraph in the main article). At the same time, separation repeatability and long term stability of untwisted Zr-BTB-FA coated column was checked. “Furthermore, identical separation performance was obtained in the separation of same isomers with 10-times injections (Supplementary Fig. 33), indicating its good column repeatability. The lifetime of the column was at least 16 months without the loss of separation efficiency (Supplementary Fig. 34), showing its feasibility for the potential applications.” has now been added (Please see Page 6, right column, 2nd paragraph in the main article)

Second, Thanks reviewer for pointing out this mistake, not “flavored”. To clearly illustrate the relationship between three main twisted conformations and untwisted structures, we have now changed “thermodynamic favored conformations” to “representative conformations”. (Please see Page 1, abstract; Page 2, right column, 1st paragraph in the main article)

Third, the stability of three representative conformations in the twisted sample was also checked under heating at 250 °C. The Moiré patterns were well kept, indicating that the three main conformations are stable in this condition. At the same time, we applied modelling to get supercell of two adjacent nanolayers with different angles. The simulations at molecular level for different angles were well-matched with their Moiré patterns, respectively. The optimized structures indicated that they are stable possibly due to the disaligned hydrogen bonding and π - π stacking. “The simulations in both models for three different angles were all well-matched with their Moiré patterns, respectively..... At the same time, the three main conformations were still stable after 250 °C treatment possibly due to the disaligned hydrogen bonding and π - π stacking (Supplementary Fig.5a-c).” has now been added. (Please see Page 2, right column, 1st paragraph in the main article).

Fourth, the transformation from twisted to untwisted Zr-BTB-FA has been investigated by H^1 NMR and DRIFTS measurements. “Considering the coordination of Zr_6 cluster, the molar ratios of $n(Zr_6 \text{ cluster}): n(HCOOH): n(BTB)=1: 2: 2$ for twisted Zr-BTB-FA as well as $1: 0.5: 2$ for untwisted Zr-BTB-FA were obtained in the H^1 NMR experiment (Supplementary Fig. 9). It indicated the 1.5 amount of formic acid loss on every Zr_6 cluster during the process of preheating, in agreement with our TGA data. This significant loss of formic acid under low temperature was probably due to the formation of ethoxy group in the ethanol mediated reactions, which recently were revealed by other groups.” has now been added. (Please see Page 5, left column, 4th paragraph in the main article). In the DRIFTS measurements, we observed the decrease of ethoxy groups in untwisted Zr-BTB-FA nanosheets, indicating the further loss of ethoxy group during the heating procedures to form possibly the chemical linkages of Zr-O-Zr between Zr-clusters in the adjacent layers through the ethoxy group-mediated linkage reaction.

5. *The authors hypothesize that their “preheating strategy” (80 °C for 6 h under*

vacuum) led to the desirable (and hopefully stable) untwisted conformations, since they believe the eclipsed conformation between adjacent layers were observed because the van der Waals interactions between each layers were too weak to prohibit the rotations in the heating-driven restacking procedures. They suggest the eclipsed structure was thermodynamically favored due to interactions (which were not clearly defined) that were stronger than van der Waals forces. It is not clear to this reviewer what the claimed interactions were or why the resultant conformations would remain stable for an extended period. The authors note that they exposed the untwisted 2-D Zr-BTB-FA nanosheets coated capillary column to 250 °C (Supplementary Fig. 22 and 23). It was not clear to this reviewer if they showed the SAME PERFORMANCE before and after this exposure. If they did this explicit check and would include a figure like that in Fig. 4 after the exposure, I would agree that the work is worthy of publication.

Response: Thank you for your comment! We have investigated interactions during the transformation from twisted to untwisted Zr-BTB-FA nanosheets by H^1 NMR and DRIFTS measurements. FA molecules were removed after heating under vacuum in the transformation from twisted nanosheets to untwisted nanosheets. Furthermore, from the H^1 NMR data, we could quantify FA in both samples. “Considering the coordination of Zr_6 cluster, the molar ratios of $n(Zr_6 \text{ cluster}): n(HCOOH): n(BTB)=1: 2: 2$ for twisted Zr-BTB-FA as well as $1: 0.5: 2$ for untwisted Zr-BTB-FA were obtained in the H^1 NMR experiment (Supplementary Fig. 9). It indicated the 1.5 amount of formic acid loss on every Zr_6 cluster during the process of preheating, in agreement with our TGA data. This significant loss of formic acid under low temperature was probably due to the formation of ethoxy group in the ethanol mediated reactions, which recently were revealed by other groups.” has now been added. (Please see Page 5, left column, 4th paragraph in the main article).

In the DRIFTS measurements, we observed the decrease of ethoxy groups in untwisted Zr-BTB-FA nanosheets, indicating the further loss of ethoxy group during the heating procedures to form possibly the chemical linkages of Zr-O-Zr between Zr-clusters in the adjacent layers through the ethoxy group-mediated linkage reaction.

We performed control experiment to elucidate the key factor in the transformation from twisted to untwisted nanosheets. The heating of twisted structure without ethanol at 250 °C still gave twisted structures but with less FA. “It was worth noting ethanol pretreatment is essential for the transformation from twisted to untwisted nanosheets in the heating/vacuum procedure. The heating of twisted nanosheets without ethanol at 250 °C still showed twisted structures (Supplementary Fig.3), although this treatment gave 1.4 amount of formic acid loss on every Zr_6 cluster with the molar ratios of $n(Zr_6 \text{ cluster}): n(HCOOH): n(BTB)=1: 0.6: 2$ (Supplementary Fig.10). This control experiment demonstrated ethoxy group-mediated linkage reaction other than the loss of formic acid was the key to form untwisted structures.” has now been added. (Please see Page 5, right column, 1st paragraph in the main article).

Meanwhile, we provided additional GC separation on the untwisted Zr-BTB-FA nanosheets column. On the one hand, untwisted Zr-BTB-FA nanosheets coated column thermal stability were analyzed by separations of the same sample of chlorotoluene isomers 10 consecutive times after the column was conditioned up to 250 °C for 30 min,

The peak shape of every isomer has no significant change from chromatograms. The results suggested that untwisted Zr-BTB-FA column can operate in the given high temperature range with good separation performance. On the other hand, lifetime of untwisted Zr-BTB-FA nanosheets coated column were performed for separation of dichlorobenzene isomers under the same condition within 16 months, the peak shapes also did not changed. “Furthermore, identical separation performance was obtained in the separation of same isomers with 10-times injections (Supplementary Fig. 33), indicating its good column repeatability. The lifetime of the column was at least 16 months without the loss of separation efficiency (Supplementary Fig. 34), showing its feasibility for the potential applications.” has now been added. (Please see Page 6, right column, 2nd paragraph in the main article).

Supplementary Figure 33. The thermal stability and repeatability of 2-D untwisted Zr-BTB-FA capillary column for separation of chlorotoluene isomers 10 times continuously. After every separation experiment, the column was heated to 250 °C for 30 min.

Supplementary Figure 34. The lifetime of 2-D untwisted Zr-BTB-FA capillary column for separation of dichlorobenzene isomers under the same experimental condition (a temperature program of 180 °C for 1 min, and then 20 °C·min⁻¹ to 250 °C) within 16 months. The separation efficiency and peak shape did not change. The drift of retention time was attributed to shortening of capillary column length due to the unavoidable depletion.

6. *Although it is a cosmetic issue, there are many typos and English issues in the manuscript, and these should be remedied by having a native English speaker read the paper.*

Response: Thank you very much for pointing this out. We have corrected the word of "formic acid", "van der Waals interactions", "angle". (Please see Page 7, right column, 2nd paragraph; Page 3, left column, 1st paragraph; Page 2, right column 1st paragraph in the main article).

Reviewers' Comments

Reviewer #2 (Remarks to the Author):

All my concerns were well and clearly clarified and the required tests were made. The new results made the general conclusion more evident and convincing. I recommend this paper to be accepted as it is.

Reviewer #3 (Remarks to the Author):

1. I think the authors have addressed my main concern, if I understand Fig. 4d correctly in their revised manuscript. Specifically, I believe the authors are showing that their untwisted Zr-BTB coated column SIGNIFICANTLY OUTPERFORMS the HP-5MS and VF-WAXMS commercial columns. This fact notwithstanding, a check the web also refers to Metal–Organic Framework MIL-101 for High-Resolution Gas-Chromatographic Separation of Xylene Isomers and Ethylbenzene, DOI: 10.1002/anie.200906560 (by one of the authors of this paper). Moreover, another paper by other authors, Exploring reverse shape selectivity and molecular sieving effect of metal-organic framework UIO-66 coated capillary column for gas chromatographic separation, Journal of Chromatography A, Volume 1257, 28 September 2012, Pages 116-124, <https://doi.org/10.1016/j.chroma.2012.07.097> considers UIO-66 rather than the above two columns mention high performance for the para, meta xylene isomer separation.

Perhaps the commercial columns (HP-5MS and VF-WAXMS) actually are based on either MIL-101 or UIO-66, but if they are not, the authors should mention and discuss their resolution compared to these chromatographic columns. This issue needs to be handled properly before the paper is accepted.

2. A less serious, but still confusing issue should also be addressed in the final paper before it is accepted. Specifically, Fig. 5 seems to show that the interlayer distance (and hence sorption capacity) of the untwisted structure in Fig. 5d is less open; however, Fig. 3d clearly shows a much higher N₂ adsorption capacity for the untwisted structure vs. the twisted structure. The authors should clarify why Fig. 5d and Fig. 3d seem to be inconsistent.

If the authors address these two issues, I would recommend the work for publication in Nature Communications

Response to Reviewers' Comments

Reviewer #2 (Remarks to the Author):

Comments:

All my concerns were well and clearly clarified and the required tests were made. The new results made the general conclusion more evident and convincing. I recommend this paper to be accepted as it is.

Response: We highly appreciate the review for the supportive comments.

Reviewer #3 (Remarks to the Author):

Comments:

1. I think the authors have addressed my main concern, if I understand Fig. 4d correctly in their revised manuscript. Specifically, I believe the authors are showing that their untwisted Zr-BTB coated column SIGNIFICANTLY OUTPERFORMS the HP-5MS and VF-WAXMS commercial columns. This fact notwithstanding, a check the web also refers to Metal–Organic Framework MIL-101 for High-Resolution Gas-Chromatographic Separation of Xylene Isomers and Ethylbenzene, DOI: 10.1002/anie.200906560 (by one of the authors of this paper). Moreover, another paper by other authors, Exploring reverse shape selectivity and molecular sieving effect of metal-organic framework UiO-66 coated capillary column for gas chromatographic separation, Journal of Chromatography A, Volume 1257, 28 September 2012, Pages 116-124, <https://doi.org/10.1016/j.chroma.2012.07.097> considers UiO-66 rather than the above two columns mention high performance for the para, meta xylene isomer separation. Perhaps the commercial columns (HP-5MS and VF-WAXMS) actually are based on either MIL-101 or UiO-66, but if they are not, the authors should mention and discuss their resolution compared to these chromatographic columns. This issue needs to be handled properly before the paper is accepted.

Response: Thank you very much for your comments, especially the concern on our previous research. The current untwisted Zr-BTB nanosheets demonstrated much higher separation performance than previous 3-D MOF, MIL-101 and UiO-66. We have now added “which was superior to commercial capillary column with different polarity like HP-5MS and VF-WAXMS columns (Fig. 4d, Supplementary Fig. 31,32 and Supplementary Table 1) and 3-D MOF columns, MIL-101 ($R_s=1.25$) and UiO-66 ($R_s=1.00$).” (Please see **Page 6, left column, 2nd paragraph** in the main article).

2. A less serious, but still confusing issue should also be addressed in the final paper before it is accepted. Specifically, Fig. 5 seems to show that the interlayer distance (and hence sorption capacity) of the untwisted structure in Fig. 5d is less open; however, Fig. 3d clearly shows a much higher N₂ adsorption capacity for the untwisted structure vs. the twisted structure. The authors should clarify why Fig. 5d and Fig. 3d seem to be inconsistent. If the authors address these two issues, I would recommend the work for publication in Nature Communications.

Response: Thank you very much for comments. Although the interlayer distance

of untwisted Zr-BTB nanosheets is indeed shorter than the twisted Zr-BTB nanosheets in the horizontal view, the vertical view indicated that the pores of untwisted Zr-BTB nanosheets are more accessible. We have now added “In the horizontal view of the model, the interlayer distance of the untwisted nanosheets is shorter than the twisted ones (Supplementary Fig. 5). However, it was worth noting that the pores in vertical views of untwisted nanosheets are highly ordered and more accessible, while not all cavities between adjacent layers in the twisted stackings were effective, giving the different BET surface areas and pore volumes between untwisted and twisted nanosheets.” (Please see Page 3, right column, 2nd paragraph in the main article).

Reviewers' Comments

Reviewer #3 (Remarks to the Author):

I believe the authors have now addressed all of my reviews and I support accepting it for publication.

Response to Reviewers' Comments

Reviewer #3 (Remarks to the Author):

Comments:

I believe the authors have now addressed all of my reviews and I support accepting it for publication.

Response: We highly appreciate the review for the supportive comments.